# Interaction-Based Behavioral Analysis of Twitter Social Network Accounts

**Hafzullah İş [1] and Taner Tuncer [2],***

[1] Department of Computer Engineering, Batman University, 72100 Batman, Turkey; Hafzullah.Is@batman.edu.tr

[2] Department of Computer Engineering, Firat University, 23119 Elazig, Turkey

* Correspondence: ttuncer@firat.edu.tr

**Abstract:** This article considers methodological approaches to determine and prevent social media manipulation specific to Twitter. Behavioral analyses of Twitter users were performed by using their profile structures and interaction types, and Twitter users were classified according to their effect size values by determining their asset values. User profiles were classified into three different categories, namely popular-active, observer-passive, and spam-bot-malicious by using k-nearest neighbor (K-NN), support vector machine (SVM), and artificial neural network (ANN) algorithms. For classification, the study used the basic characteristics of users, such as density, centralization, and diameter, as well as suggested time series such as the simple moving average and cumulative moving average. The highest accuracy was obtained by the K-NN algorithm. The results obtained with K-NN for all classes were higher than the F1-Score values obtained for the other algorithms. According to the results obtained, classification accuracy values were found to reach a maximum of 96.81% and a minimum of 92.33%. Our classification results showed that the proposed method was satisfactory for popular-active, observer-passive, and spam-bot-malicious account separation.

**Keywords:** online social media; classification; interaction; behavioral analysis

---

## 1. Introduction

In today's world, where internet access is widespread, hardware can be integrated into micro-level devices and software is inclusive at the macro level, considerable changes have occurred in individual and social communication environments. As a result of increased opportunities and tools for internet access—due to the development of information technologies—the number of social media users has started increasing exponentially, thereby changing communication platforms, socialization areas, and topics [1,2]. his change causes highly effective transformations, especially in economic, political, and social areas such as the sudden rise/fall in stock exchange and the determination of government policies. Social networks can be used in numerous areas, such as friends, environment, knowledge acquisition, news, events, as well as announcement follow-up, marketing, and advertising. People prefer to use social media to make contact with their environment and obtain faster and more effective returns; similarly, companies use social media to make contact with their customers, institutions, and organizations within their target audiences. Social networks offer people various opportunities to share their ideas and thoughts instantaneously, to make news and announcements, and to make the advertising and marketing of their products, services, and brands versatile, without requiring any intermediaries.

Twitter is a popular social media platform that is used both as a social network and a microblog. Twitter users interact with others through short texts called tweets. In addition to official institutions and organizations all around the world, twitter is actively used by heads of state, ministers, party

leaders, significant political figures, celebrities, athletes, and trademarks. Twitter is the most prestigious microblog due to its corporate identity and to the effective self-expression and interaction opportunities it allows. It played a determining role in social events such as the Arab Spring, where demonstrators organized and made their voices heard [3]. World leaders express their feelings and thoughts about important situations by means of Twitter. A tweet can raise the stock market, make governments fall, or create social movements followed by millions of people. Twitter is consistently dynamic, up-to-date, and versatile in terms of political, social, and economic aspects, and it includes rich content supplied from different sources. Therefore, by considering user profiles, the analysis of the data obtained can be used to obtain value-added information. The processing of data obtained from social network resources aids the discovery of valuable knowledge. On the other hand, it is important—in terms of accurate analysis and accurate results—to verify these accumulated data, in order to ignore or prevent interactions arising from fake profiles and use real interactions sourced from real people. Interactions such as membership information, friendship relations, shares, likes, and the comments of social network users are valuable data. Social network analyses on these scalable data can reveal relationship structures, interactions, aggregations in the network, profile asset values, and confidence indices based on the behavioral analysis of users [4]. Consistent, real, and comprehensive data is essential for effective knowledge discovery. Consumers want to review comments, likes, or complaints of other users before purchasing products and services [5]. At this point, the following are important examples of creating fake interactions. It was determined that interactions such as comments, likes, and product views made on e-commerce sites such as Amazon and Alibaba, as well as one-third of the comments made on Trivago, were misleading and obtained from fake profiles [6]. The same situation is repeated frequently for brands, people, situations, and products through fake and bot accounts on social media. The main goal of spam-bot-malicious accounts is to influence users' opinions and thoughts about politics, products, and services on social media. By using fake profiles, various actions such as distributing disturbing content, false news and comments, misleading surveys, and attempting to damage reputations or create a false reputation is possible [7]. Fake profiles can manipulate ideas and cause misleading results. Directing user habits, as well as creating interactions for changing thoughts, can misdirect ideas and thoughts.

Social networks have become the target of attackers and abusers because social media has reached billions of users, and the interactions on these platforms have significant effects on economic, political, and social interests. The social networking and microblog site Twitter, which has gained popularity all around the world in the last decade, has become the platform for cyber criminals to send spam, spread malicious messages, send phishing links, add fake profiles to the network, and participate in other malicious activities. The falsification of information and manipulation on these platforms using fake, bot, and malicious accounts has intensified in recent years.

Social networks are not successful enough in preventing spam-bot-malicious accounts, although they have their own detection algorithms. Several methods have been proposed in academic studies to solve this problem and the results were partially effective [8,9]. The major disadvantages of the methods suggested in previous studies have been the use of limited metrics, the dependence of parameters on static data, and lack of effective and comprehensive metrics to scale instantaneous changes [9]. Fake profiles are generally determined by analyzing their profile information and tweets [10]. However, this can be misleading because there are numerous fake profiles in a passive state that never tweet or interact with others but are created by phishing fraud using an original profile. These accounts are created to increase the number of followers, using the jargon "If you follow me, I will follow you", and they are used for manipulation once they have reached a very large number of followers. Therefore, such fake profiles cannot be detected simply by analyzing their profile information and tweets. To eliminate such disadvantages, methodological approaches are presented in this article to detect and prevent Twitter-specific manipulation on social media. Behavioral analyses of Twitter users were performed by using their profile structures and interaction patterns. Their asset values were determined and then a method was proposed that could make a classification according to effect size

values. To achieve this, ten metrics, namely tweets, account age, follower rank, average retweets, average likes, diameter, density, reciprocity, centralization, and modularity were used. Users were classified into three different categories such that they were effective, had strong metrics, and had a comprehensive and high added value.

The rest of the study was organized as follows. The second section presents literature studies. The third section includes the methods used and the formation of the data. The experimental results are given, and the method applied is evaluated in the fourth section. The comparison of the obtained results with the literature is given in the fifth section. The conclusion section, Section 6, outlines the contribution of this study to published literature and discusses the future development of the method applied.

## 2. Related Works

There have been studies in literature on the detection of spam-bot-malicious accounts on Twitter and other social media sites [11–23]. Social bots are programs that automatically generate content, distribute it over a particular social network and interact with the users [12]. In addition to detecting fake, bot, and spam accounts in social media, it is also important to determine the interaction made by these accounts [6]. Varol et al. determined that between 9% and 15% Twitter accounts were bot accounts [7].

Spam tweets can change Twitter statistics and lead to false information sharing and false bilateral collaborations [13]. For example, it is important to determine—in terms of measuring popularity, managing and developing processes, and in evaluating investment support status—whether the source of interactions of any election campaign, brand, product, or advertising on social media is due to popular users, passive users, or spam-bot accounts [8–14].

Fake interactions made on e-commerce platforms through social network accounts can sabotage the secure shopping environment of consumers [5]. Because users care about the comments and likes of other people involved in e-commerce, interactions such as news, comments, and likes are created by using fake profiles on these platforms with Sybil attacks, and the perceptions created by these interactions can direct consumers to certain products and services [7]. The effort to create interactions with fake followers in order to gain reputation, popularity, and a strong profile perception is another manipulative issue. Politicians, in their election campaigns, are involved in interactions with fake-bot accounts to increase their popularity on social media, as are brands for product advertisements and companies for customer relations, and these actions may show themselves very differently [4]. Social bot accounts (Sybils) have become more complex and deceptive in their efforts to replicate the behavior of normal accounts. Therefore, there is a particular need for research communities to develop technologies that can detect social bots.

Another problem is that bot and fake profiles are used to increase the number of followers [9]. The number of viewers of a social media account and its size, in terms of number of followers or friends, is a good measure of its user popularity. Likes, follows, and opinions can be purchased from social media fraud services to rapidly gain artificial popularity on social networks online.

Popular social networks have been targeted more by Sybil attacks, and bot and spam manipulations due to the size of user databases and the scope of their audience [24]. Two different approaches become prominent in determining the contents of spam. These are analyses based on user profile behavior [13].

Go et al. automatically classified tweets as positive or negative according to the inquiry [23]. They performed a semantic analysis of tweets by means of their study. However, it is a disadvantage of the study that it is limited only to two classes. De Choudhury et al. developed an automatic classifier for the user types on Twitter. A method classifying Twitter users into three categories, namely organizations, newspapers/media bloggers, and ordinary individuals, was suggested [25]. Gee et al. presented a method focusing on detecting spam accounts based only on user characteristics by manually labeling the data collected from Twitter for each profile [14]. Profiles were classified with 73% accuracy by applying the naive Bayes algorithm. Because the performance was low, an 89.6%

classification performance rate was obtained in training sets with a linear support vector machine (SVM) classifier by 5-fold cross-validations. Moreover, Benevento et al. presented a study comparing two approaches based on profile analysis and behavioral analysis to determine spam profiles and tweets [15]. An 84.6% success rate was achieved with an SVM classifier. In order to categorize tweets as spam or non-spam, an 87.6% performance rate was obtained using qualification based on the user characteristics and content. Lee et al. used user- and content-based features to distinguish spam users from other Twitter users [16]. In the study, SVM, simple logistics, and decision trees were used as classifiers. Two different data sets were used in the classification: one with 10% spammers and 90% non-spammers and one with 90% spammers and 10% non-spammers. They obtained an 88.98% classification performance rate.

Beside these studies, individual profiles, diffusive behaviors, message contents, and social relations were also considered to identify spammers [17–19]. Chen et al. focused on the "Twitter Spam Shift" problem where spammers have similar semantics but send different types of tweets. They proposed an "Lfun" approach that addresses the "Twitter Spam Shift" problem using untagged tweets [20]. The weakness of this approach is that it accepts social networks as a static system, whereas spam resources constantly develop new methods. Flora et al. studied whether the focal point in investigating phishing (identity fraud) performed using spam, bot, and malicious accounts was the detection of anomalies [21]. They stated that many anomaly detection techniques were based on behavioral patterns of normal users. Based on a two-stage detection strategy, they suggested a method for determining human behavior in a social network.

Adewole et al. proposed a similarity-based approach to identify spammers [22]. Various features were introduced to improve the performance of the three classification algorithms selected. The proposed approach was applied for a principal component analysis. The k-means algorithm was used to determine the clusters of spammers. Over 200,000 accounts selected from 2 million tweets were clustered. Experimental results showed that the random forest classifier showed the highest accuracy (96.30%). This result was followed by multi-level confirmatory factor analysis with an accuracy of 96% and the support vector machine algorithm with an accuracy of 95.69%. It was shown that the performance of selected classifiers based on class imbalance was obtained with the highest random forest algorithm in terms of accuracy, sensitivity, recall, and the F1-criterion. Researchers investigated the effectiveness of using a blacklist to reduce the amount of spam on Twitter and determined that the blacklist approach slowed down the defense against spam in social networks [20]. Vorakitphan et al. found that spammers use the click bait strategy to attract legitimate users' attention on social media platforms [26]. In their study, Aiyar et al. developed an N-gram supported spam comment detection model on the YouTube social media platform.

Meligy et al. proposed a detection technique called the fake profile identifier that could be used to verify and detect fake profiles on social networks [27]. The suggested method was based on the identifier, regular expression, and descriptive state machines. In order to improve the accuracy of the spam detection system, Kiliroor et al. proposed a system to detect and prevent spam based on a naive Bayes spam filtering approach [28]. In the study, social context features such as stocks, likes, and comments were used to evaluate the performance of proposed models. It was found that the proposed system achieved a higher sensitivity than the accuracy of the basic classifier. That research focused particularly on the identification of spam. Chiyu et al. proposed a model based on behavioral analysis [29]. They used a data set containing 5658 social media users. The performance was tested with four different algorithms and an 87.32% accuracy rate was obtained by the proposed method, behavior enhanced deep bot detection in social media. The proposed model considered the user content as temporary text data instead of plain text in order to extract hidden temporary patterns. Moreover, this model combined content knowledge and behavioral knowledge using a deep-learning method. Eiman et al. presented a study including new techniques designed to distinguish between social bot accounts and real user accounts [12]. They limited their analysis to the detection of social

bots on the Twitter social media platform. They reviewed various detection schemes currently in use and examined common aspects such as classifiers, data sets, and selected features.

The algorithms used to determine the profile quality and the features included in studies can vary. Different parameters were used in the methods used by the nine different studies given in Table 1. Although different algorithms use tweet counts, profile samples, profile attributes, and behavioral attributes, their performance varied between 86% and 96%. The most successful method is the approach used by Kayode et al. [22], which had a 96.30% success rate. This study was followed by the study of Chu et al. [30], which had a 96% success rate. Both profile and behavioral features were included in the study and a high success rate was obtained by using 11 different metrics in the study, including 6000 samples and 8,350,095 tweets.

**Table 1.** Metric selection and data set collection summaries for machine learning.

| Ref.# | ∑ # Tweets | ∑ # Sample | Extracted Features | Features * P/B |
|-------|-----------|-----------|--------------------|----------------|
| [10] | 8,350,095 | 6000 | 11 | P + B |
| [22] | 2M | 200K | - | B |
| [29] | - | 1000 | 16 | P + B |
| [31] | 722,109 | 3536 | 15 | P + B |
| [32] | - | 3020 | | B |
| [33] | 5.6M | 15K | - | P + B |
| [34] | - | 9134 | - | P + B |
| [35] | 5.6M | 31K | 1000 | P + B |

\* P: profile features; B: behavioral features.

## 3. Data and Method

The proposed process to classify users according to metric data is given in Figure 1. Three different groups are defined such that social media users—the focus of this study—will be labeled in order to perform their behavioral analyses based on physical interaction. These groups are popular-active, observer-passive, and spam-bot-malicious. In general, these categories are the most optimum combinations in which social media users can be grouped in accordance with the literature studies and experimental research.

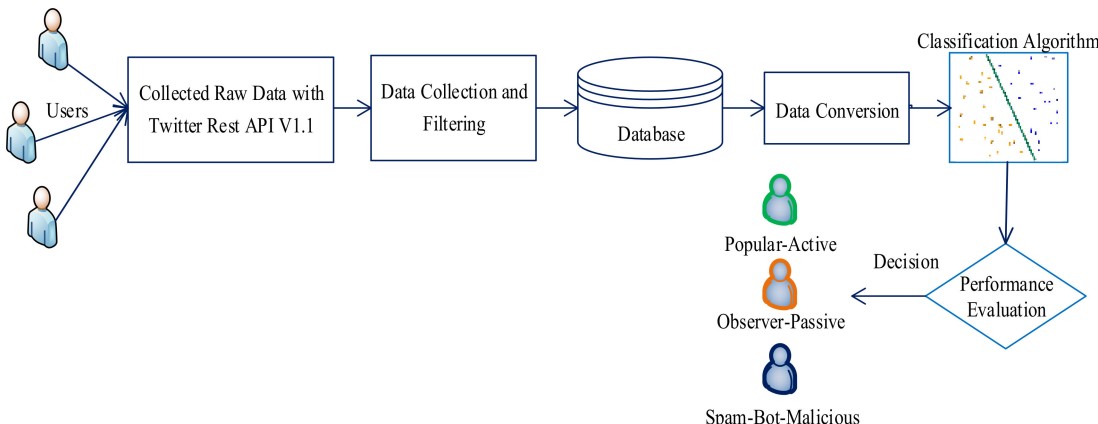

**Figure 1.** Flow diagram of the study.

### 3.1. Data Set

Literature studies usually use data sets consisting of data collected from many different metrics and user counts collected over different time periods. The data set used in this article was formed using

completely up-to-date data obtained as a result of metrics measured in terms of scope and efficiency, sufficient and effective user counts, and filtering processes.

*3.2. Data Collection and Filtering*

To classify users correctly and make the classification performance high—in addition to parameters used in the literature such as tweets, account age, follower rank, average retweets, and average likes—other parameters such as diameter, density, reciprocity, centralization, and modularity were used. These metrics are the parameters that focus on a different area to reveal many aspects in which social network users interact. The data used to create the data set was collected from Twitter. The metric data forming the data set was extracted using Twitter Rest API V1.1 supporting search/tweet endpoints by means of the SocialBlade and Netlytic platforms. The frequency at which Rest API can extract Twitter data is at 15 min intervals. It allows the return of a maximum of 1000 records each time.

The data set, consisting of the parameters mentioned above, had data extracted at two-week intervals for each user over an eight-month period. The data of 4209 users were extracted to be used as training and test data. Of these users, the data of 4014 users were used as training, and the remaining 195 data were used for test data. Among the users in the training set, 1791 people were in the popular-active group, 891 were in the observer-passive group, and 1332 were in the spam-bot-malicious accounts group. Of the test set users, 84 were in the popular-active group, 42 were in the observer-passive group, and 69 were in the spam-bot-malicious group. The data were extracted from Twitter between September 2018 and April 2019. At each periodic scan, a maximum of 1000 of the most recently shared and updated parameters were included in the data set. The total number of tweets with measured interaction was 2,820,030. The data set, which was formed by calculating ten different metrics from 4209 users over a monthly period, consisted of 673,440 metric data. The size of sixteen periods of data was 5120 KB. Table 2 shows the characteristics of the data set.

**Table 2.** Qualities of the data set with label and number of profiles used.

| Label | Quality of Group | Training | Test | Total |
|-------|------------------|----------|------|-------|
| 0 | Popular-active users | 1791 | 84 | 1875 |
| 1 | Observer-passive users | 891 | 42 | 933 |
| 2 | Spam-bot-malicious users | 1332 | 69 | 1401 |
| | Total User Counts in Data Set | | | 4209 |
| | Data Acquisition Process | | | 8 months 16 weeks |
| | Total Number of Metrics | | | 673,440 |
| | Total Data Size | | | 5120 KB |
| | Classification Categories | | | 3 |
| | Tweet Counts with Measured Interaction | | | 2,820,030 |

The explanations and purposes of the use of metrics in the data set are given below:

(a)  Tweets *(TW)*: They are the messages sent by users to communicate with others. These messages are the most effective way to create interaction. The tweet activity of the user presents significant information about his or her character. It is the most effective scale used to detect observer-passive users in particular.

(b)  Account Age *(AA)*: This refers to time elapsed between the opening of an account and its active time. It is one of the basic criteria for detecting spam-bot-malicious accounts.

(c)　Follower Rank *(FR)*: This is the ratio of the number of following accounts *(FD)* to the number of follower accounts (FW). *FR* is calculated as shown in Equation (1) below:

$$FR = \frac{FD}{FW}.$$ 　　　　　　　　(1)

The popularity of the account increases as this ratio decreases. Popular users have many followers, but they follow few accounts. The opposite is observed in spam-bot-malicious accounts. They have very high number of following accounts, but their follower accounts are very low.

(d)　Average Retweet *(AR)*: It refers to a case where a user shares the message of another user in his/her profile. The average retweet is high in popular-active and observer-passive accounts, and it is low in spam-bot-malicious accounts.

(e)　Average Likes *(AL)*: This parameter expresses that a social media user likes the messages shared by others. It is very high for popular-active social media users, but low for observer-passive and spam-bot-malicious accounts.

(f)　Density *(DN)*: This is the ratio of current connections to the total number of possible connections in a network. This parameter helps to show the closeness of participants within a network. The closer this measurement is to 1, the more interaction participants have with each other. On the other hand, the fact that the density is close to 0 indicates a lack of communication and lack of interaction. Assuming that $N$ is the number of nodes, $C$ is the number of connections, and P is the number of all possible connections, the number of possible connections in the network is calculated by Equation (2) and density is found by Equation (3):

$$C = \frac{N * (N-1)}{2},$$ 　　　　　　　　(2)

$$DN = \frac{C}{P}.$$ 　　　　　　　　(3)

(g)　Centralization *(CZ)*: This measures the average centralization degree of all nodes within a network. When a network has a high centralization value close to 1, this indicates that there are several central participants dominating the information flow in the network. In the networks with a low centralization measurement close to 0, it appears that information flows more freely between most participants, i.e., centralization—that expresses dependency—is very low.

In dynamic networks, the dynamic centralization metric is used for calculating centralization of a node, as shown in Equation (4). In a graph such as $G = (V,E)$, assuming $V$ is the edge and *(s,t)* is the shortest path for each edge pair, the fraction of shortest paths passing the peak point is determined for each edge pair. This fraction is summed at all pairs of peak points. In Equation (4), σst is the sum of shortest paths between node $s$ and node $t$, and σst(v) is the number of shortest paths passing the edge V:

$$CZ = \sum_{s:=v:=t \in V} \frac{\sigma st(v)}{\sigma st}.$$ 　　　　　　　　(4)

Determining the structural connection of nodes provides important information about the structural flow of the data flowing through them. Normalized degree centralization (CZ$_{ND}$) is calculated as shown in Equation (5):

$$CZ_{ND} = di * (N-1).$$ 　　　　　　　　(5)

(h)　Reciprocity *(RC)*: This is a ratio of connections that indicates two-way communication (also called reciprocal connection) in relation to the total number of current connections. A high parameter level indicates that many participants have two-way conversations, while a low parameter level

means that many conversations are one-sided. This parameter is important in revealing the interaction of a node is structured. Assuming that the number of links with two-way connections is $L< - >$ and the number of all active connections is $L$, reciprocity is calculated as shown in Equation (6):

$$RC = L^{<->}/L. \tag{6}$$

(i)  Diameter *(DT):* This calculates the longest distance between two network participants. This value indicates the size of a network by calculating the number of nodes required for two users to reach each other. If the diameter increases, the domains of individuals increase and their interaction potential also increases.

(j)  Modularity *(MD):* This determines whether clusters represent different communities in the network. Higher values of modularity are represented by clusters and indicate net divisions between communities. In a G *(U, V)* network, assuming *V* is a node and *E* is the edge, $e_{ii}$ indicated in Equation (7) is the edge percentage and $a_i$ indicated in Equation (8) is the percentage of edges terminated on at least one side. *MD* is calculated by Equation (9).

$$e_{ii} = \left|\{(u,v) : u \in V_i,\ v \in V_i,\ (u,v) \in \mathrm{E}\}\right|/|\mathrm{E}|, \tag{7}$$

$$a_i = \left|\{(u,v) : u \in V_i,\ (u,v) \in \mathrm{E}\}\right|/|\mathrm{E}|, \tag{8}$$

$$MD = \sum_{i=1}^{c}\left(e_{ii} - a_i^2\right). \tag{9}$$

In order to make the data set usable, various processes such as data cleaning, data integration, data reduction, and data conversion should be applied. Time series were used to extract all outlier, inconsistent and inappropriate data, and to complete missing data. In the data set, time series were applied to a sixteen-period data set and the dimension of the data was reduced to 1D. Data reduction operations were separately performed by using the simple moving average (SMA) and cumulative moving average (CMA). This data set was used because the data reduced to 1D were found to be more consistent and meaningful with the SMA. The equations of SMA and CMA are given in Equations (10) and (11), respectively. A complete, noise-free and consistent data set was created through the effective solution of time series. Table 3 shows the structure of the data set obtained as a result of dimension reduction.

$$SMA = \frac{Pm + Pm - 1 + \cdots + P(m - (n - 1))}{n}, \tag{10}$$

$$CMA = \frac{X1 + \cdots + Xn}{n}. \tag{11}$$

**Table 3.** The database formation stages of the data set (P: Period, U: User, and M: Metric).

| Stage 1. Collection of Data in 16 Time Periods | | | | | | | | | |
|---|---|---|---|---|---|---|---|---|---|
| Users | P1 | P2 | P3 | P4 | P5 | P6 | P7 | P8 | ... | P16 |
| U1 | M1–10 | M1–10 | M1–10 | M1–10 | M1–10 | M1–10 | M1–10 | M1–10 | M1–10 | M1–10 |
| ... | M1–10 | M1–10 | M1–10 | M1–10 | M1–10 | M1–10 | M1–10 | M1–10 | M1–10 | M1–10 |
| U4209 | M1–10 | M1–10 | M1–10 | M1–10 | M1–10 | M1–10 | M1–10 | M1–10 | M1–10 | M1–10 |
| Stage 2. Dimension Reduction on Data Using Time Series | | | | | | | | | |
| | M1 | M2 | M3 | M4 | M5 | M6 | M7 | M8 | M9 | M10 |
| U1 | D1 | D2 | D3 | D4 | D5 | D6 | D7 | D8 | D9 | D10 |
| ... | ... | ... | ... | ... | ... | ... | ... | ... | ... | ... |
| U4209 | D1 | D2 | D3 | D4 | D5 | D6 | D7 | D8 | D9 | D10 |

M: Metric, P: Period, U: User, and D: Data obtained as a result of dimension reduction.

### 3.3. Data Conversion

Working on a data set of raw data extracted from Twitter Application Programming Interface (API)s can often lead to inconsistent results. Each metric has a value in different ranges. Mean and variance values for metrics can be in different ranges with respect to each other. The effect of metrics with large mean and variance values on classification performance can be higher than others. Normalization was therefore performed to equalize the effect of all metrics on classification. One of the methods that can be used for normalization is min-max normalization. The min-max normalization method was used to convert all metrics into numerical values between 0 and 1. This method is based on the principle of determining the largest and smallest numerical value in the data and converting the values accordingly. The min-max normalization equation is shown in Equation (12). *X\** indicates converted values, *X* indicates observation values, $X_{min}$ is the smallest observation value, and $x_{max}$ is the highest observation value. Table 4 shows the normalized values of each metric between 0 and 1 for some data.

$$X^* = \frac{X - Xmin}{Xmax - Xmin} \tag{12}$$

**Table 4.** Data set conversions with min-max optimization. Tweets (*TW*); Account Age (*AA*); Follower Rank (FR); Average Retweet (*AR*); Average Likes (*AL*); Density (*DN*); Centralization (*CZ*); Reciprocity (*RC*); Diameter (*DT*); Modularity (*MD*).

| | | | Unfiltered Data | | |
|---|---|---|---|---|---|
| **User** | **Katy Perry** | **Justin Bieber** | **Barack Obama** | **Rihanna** | **Taylor Swift13** |
| TW | 9192.00 | 30,371.00 | 1408.00 | 10,068.00 | 78.00 |
| AA | 104 | 104 | 128 | 104 | 116 |
| FR | 1 | 2 | 3 | 4 | 5 |
| AR | 1.78 | 69.04 | 6.76 | 7865.42 | 22.54 |
| AL | 12.82 | 198.11 | 360.49 | 43.95 | 83.31 |
| DN | 20.00 | 38.00 | 2.00 | 11.00 | 3.00 |
| CZ | 0.002933 | 0.006929 | 0.003517 | 0.005980 | 0.002025 |
| RC | 0.004575 | 0.000810 | 0.000810 | 0.074400 | 0.029980 |
| DT | 0.290370 | 0.290970 | 0.239170 | 0.096340 | 0.375770 |
| MD | 0.606270 | 0.451470 | 0.627370 | 0.537670 | 0.411870 |
| | | | Min-Max Optimization Applied Data | | |
| **User** | **Katy Perry** | **Justin Bieber** | **Barack Obama** | **Rihanna** | **Taylor Swift13** |
| TW | 0.00000275 | 0.00000275 | 0.00000338 | 0.00000275 | 0.00000307 |
| AA | 0 | 0.006410256 | 0.012820513 | 0.019230769 | 0.025641026 |
| FR | 0.000254714 | 0.009862714 | 0.000965714 | 1.123631429 | 0.003219286 |
| AR | 0.001622405 | 0.025077595 | 0.045632152 | 0.005562911 | 0.01054557 |
| AL | 0.016393443 | 0.031147541 | 0.001639344 | 0.009016393 | 0.002459016 |
| DN | $1.77758 \times 10^{-5}$ | $4.19939 \times 10^{-5}$ | $2.13152 \times 10^{-5}$ | $3.62424 \times 10^{-5}$ | $1.22727 \times 10^{-5}$ |
| CZ | 0.002933 | 0.006929 | 0.003517 | 0.005980 | 0.002025 |
| RC | 0.004575 | 0.000810 | 0.000810 | 0.074400 | 0.029980 |
| DT | 0.290370 | 0.290970 | 0.239170 | 0.096340 | 0.375770 |
| MD | 0.606270 | 0.451470 | 0.627370 | 0.537670 | 0.411870 |

### 3.4. Methods

To classify the profiles in the data set into three different categories with respect to their characteristics, support vector machine (SVM), k-nearest neighbor (KNN), and artificial neural network (ANN) algorithms were used. The asset values, effect sizes, qualities, and confidence indices

of profiles are determined by the character analysis obtained from the behavioral analysis based on social media interaction.

To increase the classification performance, the training set data of algorithms were kept high. Four thousand and fourteen account data were used in the training of the algorithms. Algorithms were implemented on the MATLAB platform, and tested to find out the appropriate parameters for the highest performances with 5-fold cross-validations applied to all classification algorithms. Confusion matrix values of all the applied algorithms were calculated and their receiver operating characteristic (ROC) curves were plotted.

A confusion matrix is a popular representation of the performance of classification models. The matrix shows the number of correctly and falsely classified samples compared to the actual results with the test data. One of the advantages of using the confusion matrix as an evaluation tool is that if the data set is imbalanced, it allows a more detailed analysis rather than just a simple proportion of correctly classified samples, which can produce misleading results. The matrix dimensions are $n \times n$, where $n$ indicates the number of classes [36]. Table 5 shows the confusion matrix for 2 classes.

**Table 5.** Confusion matrix.

|  | **Actual Positive** | **Actual Negative** |
|---|---|---|
| Predictive Positive | True Positive (TP) | False Positive (FP) |
| Predictive Negative | False Negative (FN) | True Negative (TN) |

The performance parameters for the algorithms were calculated by using true positive (*TP*), false positive (*FP*), true negative (*TN*), and false negative (*FN*) values in the confusion matrix. The calculated parameters are Accuracy, Precision, Recall and F1-Score.

Accuracy is the ratio of accurate estimates to all estimates in the system:

$$Accuracy = \frac{(TP + TN)}{(TP + TN + FN + FP)}. \tag{13}$$

Recall indicates how successful positive situations have been estimated:

$$Recall = \frac{TP}{(TP + FN)}. \tag{14}$$

Precision indicates the success for a situation estimated as positive:

$$Precision = \frac{TP}{(TP + FP)}. \tag{15}$$

F-Measure is the harmonic average of recall and precision, and it indicates the overall accuracy of classification:

$$F - Measure = \frac{2 * Precision * Recall}{(Precision + Recall)}. \tag{16}$$

ROC curves, one of the performance measures in classification, are performance measures for the classification problem at various threshold values. ROC is a probability curve and the area under the curve (AUC) represents the degree or measure of separability. The higher the AUC is, the higher the performance in distinguishing the class of profiles will be. In a perfect model, proximity to 1 means a good separation measure. In a weak model, proximity to 0 means the worst measure of separability in the AUC. When the AUC is 0.5, it means that the model has no class separation capability [37].

## 4. Experimental Results

In this study, behavioral analyses of social media users were performed, and they were classified according to the characteristics and interactions of their accounts. An up-to-date data set consisting of 4209 training and test users was created. Ten different metrics were used in the data set, which could effectively scan the qualities and interactions of each user in their profiles. Data were collected for sixteen periods, or twice a month for eight months. In the data set, simple and cumulative average methods were used with time series for dimension reduction. The variance between the data was corrected using the min-max normalization method.

A confusion matrix was used for the performance measurement of SVM, K-NN, and ANN classification algorithms. Figure 2 shows the confusion matrix results obtained for each algorithm.

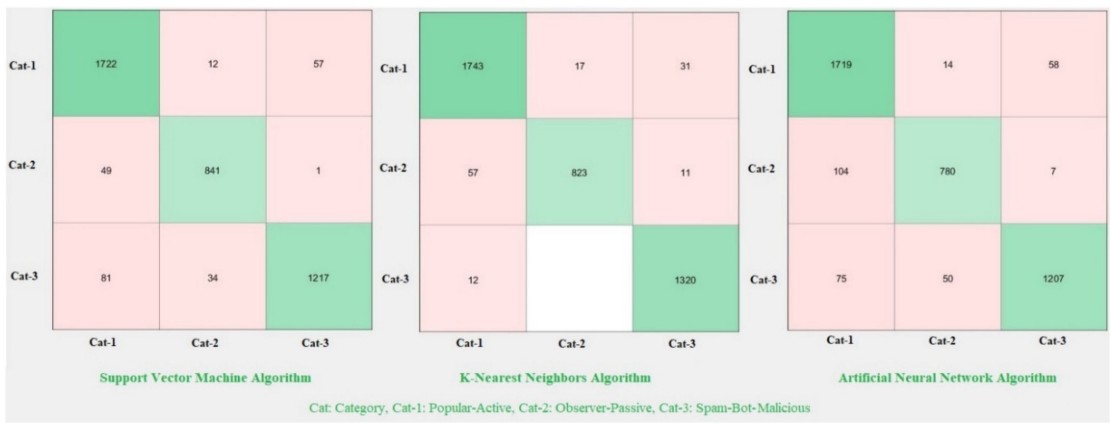

**Figure 2.** Confusion matrix results.

Examining the confusion matrices, the detection accuracy of the algorithms in the classification is close to each other and their performance is quite high. Success rates, accuracy, precision, recall, and F1-Score values for each classification algorithm were calculated and are presented in Table 6.

**Table 6.** Accuracy, precision, recall, F1-Score values for the classification algorithms.

| Category | n(Truth) | N(Classified) | Precision | Recall | F1-Score | Accuracy | Overall Accuracy |
|---|---|---|---|---|---|---|---|
| **Support Vector Machine Algorithm** | | | | | | | |
| Popular-Active | 1852 | 1791 | 0.96 | 0.93 | 0.95 | 95.04% | |
| Observer-Passive | 887 | 891 | 0.94 | 0.95 | 0.95 | 97.61% | 94.17% |
| Spam-Bot-Malicious | 1275 | 1332 | 0.91 | 0.95 | 0.93 | 95.69% | |
| **K-Nearest Neighbors Algorithm** | | | | | | | |
| Popular-Active | 1812 | 1791 | 0.97 | 0.96 | 0.97 | 97.09% | |
| Observer-Passive | 840 | 891 | 0.92 | 0.98 | 0.95 | 97.88% | 96.81% |
| Spam-Bot-Malicious | 1362 | 1332 | 0.99 | 0.97 | 0.98 | 98.65% | |
| **Artificial Neural Network Algorithm** | | | | | | | |
| Popular-Active | 1898 | 1791 | 0.96 | 0.91 | 0.93 | 93.75% | |
| Observer-Passive | 844 | 891 | 0.88 | 0.92 | 0.90 | 95.64% | 92.33% |
| Spam-Bot-Malicious | 1272 | 1332 | 0.91 | 0.95 | 0.93 | 95.27% | |

According to the classification results, the average accuracy value is 94.17% for the SVM algorithm, 96.81% for the K-NN algorithm, and 92.23% for the ANN algorithm. In the K-NN algorithm, showing

the highest overall accuracy value, the accuracy value is 97.09% for the popular-active class, 97.88% for the observer-passive class, and 98.65% for the spam-bot-malicious class. Examining the F1-Score parameter, which is the harmonic average of precision and sensitivity, and which indicates the overall accuracy of the classification, the highest accuracy was obtained by the K-NN algorithm. The results obtained with K-NN for all classes are higher than the F1-Score values obtained for the other algorithms.

ROC curves, another indicator of performance measurement in classification, were obtained for each algorithm. Examining the ROC curves, the AUC value of the K-NN algorithm is close to 1. The ROC curve shows that the success of the K-NN algorithm in predicting the class of profiles is clearly distinguishable from other algorithms. SVM and ANN algorithms had AUC values of 0.99 and 0.98, respectively, while the highest value (1.0) was obtained with the K-NN algorithm.

The results obtained in Figure 3 show that more successful and acceptable results were obtained with the K-NN algorithm.

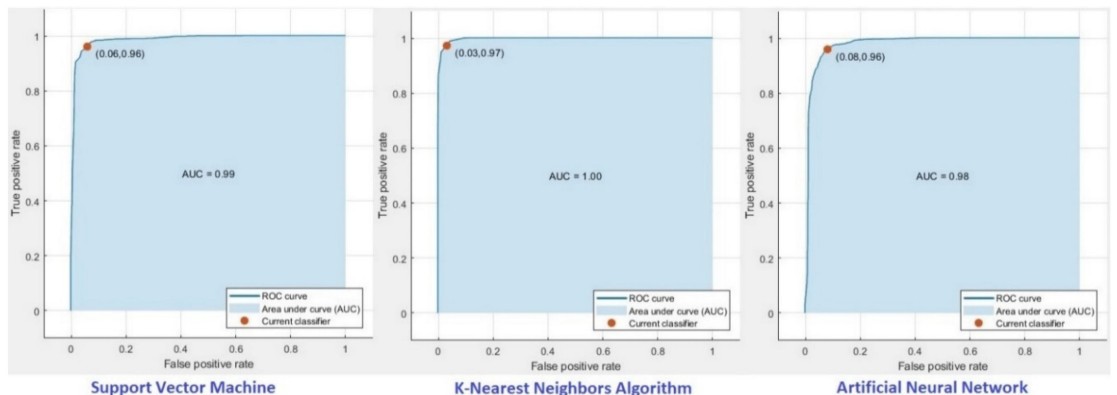

**Figure 3.** Receiver operating characteristic (ROC) curve results of the classification algorithms.

## 5. Discussion

One of the major problems in social media platforms such as Twitter is bot, Sybill, fake, spam accounts, which are controlled by automated software, which are often used for malicious activities. It is important to identify Bot, Sybill, fake, spam accounts that affect a community on a particular topic, spread false information, direct people to illegal organizations, manipulate people for stock exchange transactions, and attempt to spread their private information. The determination of bot, fake, Sybill accounts has been widely studied in the literature. The studies on the separation of bot, fake, and Sybill accounts from normal users by using machine learning methods are given in Table 7. With the use of many machine learning algorithms, normal user and bot, fake, Sybill account classification has been determined with 90%–96% accuracy. The studies in the literature suggested the use of one or more tweet features, periodic features, and user features. In this article, the separation of popular-active, observer-passive, and spam-bot-malicious groups has an average accuracy of 96.81% with the help of the SVM, K-NN, and ANN algorithms.

The contributions made to the literature in this article can be summarized as follows:

- The users were categorized into three different categories, namely, spam-bot-malicious, popular-active, and observer-passive.
- Unlike existing literature, diameter, density, reciprocity, centralization, and modularity parameters were used to determine users' interactions.
- Simple moving average and cumulative moving average were used with time series to separate all outlier, incoherent and inconvenient noisy data, and to complete missing data.
- The SVM, ANN, and K-NN classification algorithms were used and the classification accuracy values had maximum value of 96.81% and a minimum of 92.33%.

Our classification results showed that the proposed method was satisfactory for popular-active, observer-passive, and spam-bot-malicious account separation. The results determined that 10 different parameters were adequate and effective in classifying the users.

**Table 7.** Methods and success rates of studies in the literature.

| Reference | Method | Accuracy | Account Detected |
|---|---|---|---|
| [22] | Random Forest, Artificial Neural Network, and Support Vector Machine | 96.3% | Spam |
| | | 96% | |
| | | 95.6% | |
| [31] | Random Forest | 86.44% | Automated, Normal Users |
| [32] | Decision tree, Bayesian Network, Support Vector Machine, and Artificial Neural Network | ~90% | Human, Sybill |
| [33] | Random Forest | 95% | Bot, Sybill |
| [34] | Warped Correlation Method | 94% | Bot |
| [35] | Logistic Regression, Naive Bayes, Support Vector Machine, and Gradient Boosted Trees | 75% | Bot |
| | | 78% | |
| | | 82% | |
| | | 86% | |
| [29] | Naive Bayes and Entropy Minimization Discretization | 90% | Fake |
| [12] | Random Forest | 96% | Human, Bot, and Cyborg |
| [10] | Artificial Neural Networks, Decision Tree Classifier, and Random Forest Classifier | 92.08% | Fake |
| | | 94.58% | |
| | | 95% | |
| **Proposed Method** | **Support Vector Machine, K-Nearest Neighbors, and Artificial Neural Network** | **96.81%** | **Popular-Active, Observer-Passive, and Spam-Bot-Malicious** |

## 6. Conclusions

In social networks, users can be classified according to their activity and basic descriptive characteristics, and they can also be classified according to their behavior on the network. Users cannot be easily grouped into social networks like in everyday life, and their relationship patterns cannot be derived only by their sharing or descriptive qualities. The interactions and relationships of profiles with multiple communities may differ. However, studies show that there is a similarity between the individual character of a social network user and his or her behavior on social networking environments online. To determine this in the study, behavioral analyses were conducted by investigating the structure of the interaction schemes of online social media users. The classification was successfully performed in accordance with users' characteristics.

The fact that the data set used is up to date and comprehensive has been one of the main aspects in achieving an effective result with high performance. High performance rates from the most popular data mining algorithms used in classification have determined the accuracy of the methodology applied. The original, comprehensive, and innovative method presented in this study is expected to provide the basis for qualified applications in the field. Our possible future research is to use new machine learning algorithms such as deep learning to more accurately detect spam-bot-malicious accounts.

**Author Contributions:** Investigation, data collection and filtering, data conversion, software, classification, test, H.İ.; Problem definition, methodology, evaluation of obtained results, writing—review and editing, T.T.

**Conflicts of Interest:** The authors declare no conflict of interest. This research received no external funding.

**Data Availability:** Twitter data used in this article can be obtained from the link below. https://github.com/hafzullahis/mdpi-applsci-603816.

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
