# Peer review of "Interaction-Based Behavioral Analysis of Twitter Social Network Accounts"

_applsci, doi:10.3390/app9204448_

Round 1
Reviewer 1 Report
The article proposes a set of different artificial intelligent techniques in order to classify twitter users into: Popular-Active, Observer-Passive and Spam-Bot-Malicious types. To do so they use the information about the number of tweets during a period of time and other metrics related to dynamic networks.
I think that the article is well-written and structured. I think that changes made from the first version was correct. Now the discussion is clear and the results are correctly described. Table 7 is providing the information that the reader need to understand the contributions of this proposal.
Author Response
Thank you for your comments and contributions.
Reviewer 2 Report
This manuscript introduces an approach to identify spam and malicious tweets. The manuscript fits within the objectives and scope of the journal and presents a reasonable degree of scientific soundness and adequate technical details. However, I recommend the following changes to make the manuscript publishable:
1- Figure 1 on Page 6 does not reflect the experiment described. This diagram/figure should provide a graphical representation of both the steps and sequences carried out in the study.
2- The authors should consider adding a discussion section right before the conclusion to discuss the implications of their study and how it can fill the existing gaps in the field and elaborate on the potential limitations and disadvantages of the proposed approach.
3- I strongly encourage authors to have the manuscript proofread by a native English speaker before re-submission.
Author Response
We would like to sincerely thank the reviewers and editor for providing us with constructive and insightful feedbacks. The paper has been revised again and improved in the light of your remarks. In this revision, some sections have been modified to address the reviewers’ comments, with the changing parts highlighted in yellow shadowed text through the manuscript.
Sincerely yours,
The authors
This manuscript introduces an approach to identify spam and malicious tweets. The manuscript fits within the objectives and scope of the journal and presents a reasonable degree of scientific soundness and adequate technical details. However, I recommend the following changes to make the manuscript publishable:
Figure 1 on Page 6 does not reflect the experiment described. This diagram/figure should provide a graphical representation of both the steps and sequences carried out in the study.
Fig.1 has been updated as follows.
2- The authors should consider adding a discussion section right before the conclusion to discuss the implications of their study and how it can fill the existing gaps in the field and elaborate on the potential limitations and disadvantages of the proposed approach.
Discussion section has been added to the article.
Discussion
One of the major problems in social media platforms such as Twitter is bot, Sybill, fake, spam accounts, which are controlled by automated software, which are often used for malicious activities. It is important to identify Bot, Sybill, fake, spam accounts that affect a community on a particular topic, spread false information, direct people to illegal organizations, manipulate people for stock exchange transactions, and attempt to spread their private information. The determination of bot, fake, Sybill accounts has been widely studied in the literature. The studies on the separation of bot, fake and Sybill accounts from normal users by using machine learning methods are given in Table 7. With the use of many machine learning algorithms, normal user and bot, fake, Sybill account classification has been determined with 90%–96% accuracy. The studies in the literature suggested the use of one or more tweet features, periodic features and user features. In this article, the separation of Popular-Active, Observer-Passive and Spam-Bot-Malicious groups has an average accuracy of 96.81% with the help of the SVM, K-NN and ANN algorithms.
The contributions made to the literature in this article can be summarized as follows:
The users were categorized into three different categories, namely, spam-bot-malicious, popular-active, and observer-passive. Unlike existing literature, diameter, density, reciprocity, centralization and modularity parameters were used to determine users’ interactions. Simple Moving Average and Cumulative Moving Average were used with time series to separate all outlier, incoherent and inconvenient noisy data, and to complete missing data. The SVM, ANN, and K-NN classification algorithms were used and the classification accuracy values had maximum value of 96.81% and a minimum of 92.33%.
Our classification results showed that the proposed method was satisfactory for Popular-Active, Observer-Passive and Spam-Bot-Malicious account separation. The results determined that 10 different parameters were adequate and effective in classifying the users.
Table 7. Methods and Success Rates of Studies in the Literature.
|
Reference |
Method |
Accuracy |
Account detected |
|
[22] |
Random Forest, Artificial Neural Network, and Support Vector Machine |
96.3 % 96% 95.6% |
Spam |
|
[32] |
Random Forest |
86.44% |
Automated, Normal Users |
|
[33] |
Decision tree, Bayesian Network, Support Vector Machine, and Artificial Neural Network |
~90% |
Human, Sybill |
|
[34] |
Random Forest |
95% |
Bot, Sybill |
|
[35] |
Warped Correlation Method |
94% |
Bot |
|
[36] |
Logistic Regression, Naive Bayes, Support Vector Machine, and Gradient Boosted Trees |
75% 78% 82% 86% |
Bot |
|
[29] |
Naive Bayes and Entropy Minimization Discretization |
90% |
Fake |
|
[31] |
Random Forest |
96% |
Human, Bot, and Cyborg |
|
[37] |
Artificial Neural Networks, Decision Tree Classifier and Random Forest Classifier |
92.08% 94.58% 95% |
Fake |
|
Proposed Method |
Support Vector Machine, K-Nearest Neighbors and Artificial Neural Network |
96.81% |
Popular-Active, Observer-Passive and Spam-Bot-Malicious |
3- I strongly encourage authors to have the manuscript proofread by a native English speaker before re-submission.
The article was edited by MDPI edit service.

Reviewer 3 Report
Thank you for sharing your research. The study has potential but it needs a better positioning in the literature and much more attention in how the flows of ideas, results and implications are presented. Please consults the following recommendations.
Lines 9-15 in the abstract (until “this article…) can be easily eliminated or placed in the introduction. A word on the implications of the results/research could be instead introduced at the end of the abstract.
The introduction is comprised of too many general statements, with a proper backing in references. For example, lines 31- 33: the number of social media users has started increasing exponentially, thereby changing communication platforms, socializationareas and topics. Is there an official sources for this increase, a period of reference/a certain place, in what way communication platforms were changed ? etc. You may want to consult the following: Boulianne, S. (2015). Social media use and participation: A meta-analysis of current research. Information, communication & society, 18(5), 524-538.
The same applies to many other phrases:
“This change causes highly effective transformations, especially in economic, political and social areas”. What do you mean by “highly effective transformations”? Lines 36-38: “People prefer to use social media to make contact with their environment and get faster and more effective returns; similarly, companies use it to make contact with their customers, institutions and organizations within their target audiences”. Can you cite a source acknowledging these motivations for social media use or do you take it as common knowledge? Lines 47-48: the role of twitter in the Arab Spring, a source for assessing its “determining role”. See: Arafa, M., & Armstrong, C. (2016). " Facebook to Mobilize, Twitter to Coordinate Protests, and YouTube to Tell the World": New Media, Cyberactivism, and the Arab Spring. Journal of Global Initiatives: Policy, Pedagogy, Perspective, 10(1), 6.
Some phrases are so general that seem to be even a truism: lines 60-61 - Consistent, real and comprehensive data is essential for effective knowledge discovery.
At a deeper level, there is a very abrupt transition to the idea of fake news, a concept used in rather confuse manner next to “fake profiles” and “fake accounts”. Are they interchangeable?
Line 73 – typo - contributions. Listed in the current form, this subsection should be moved at the end of paper, in the discussion/conclusions part. In the introduction you may want to rephrase some of these ideas, so that you it becomes clearer how this paper addresses the gap in the literature and what are the research questions.
Line 85 – you may completely eliminate the subsection title. It is just a common paragraph at the end of the introduction that does not need a heading.
I don’t understand the logic of the references in text: there is 33 in line 103, and the directly to 38 in line 116.
Section 3 – Problem definition – is much more naturally to be integrated into the introduction.
Section 4 should be renamed so that it becomes Data and Method, or Research Methodology or anything else that skips the “Proposed”.
Line 230 – typo: construction.
Figure 2 should be converted into a table.
The conclusions should be extended in order to clearly delineate the theoretical and practical implications of the paper.
Author Response
We would like to sincerely thank the reviewers and editor for providing us with constructive and insightful feedbacks. The paper has been revised again and improved in the light of your remarks. In this revision, some sections have been modified to address the reviewers’ comments, with the changing parts highlighted in yellow shadowed text through the manuscript.
Sincerely yours,
The authors
Thank you for sharing your research. The study has potential but it needs a better positioning in the literature and much more attention in how the flows of ideas, results and implications are presented. Please consults the following recommendations.
Lines 9-15 in the abstract (until “this article…) can be easily eliminated or placed in the introduction. A word on the implications of the results/research could be instead introduced at the end of the abstract.
Abstract updated as follows.
Abstract: This article considers methodological approaches to determine and prevent social media manipulation specific to Twitter. Behavioral analyses of Twitter users were performed by using their profile structures and interaction types, and Twitter users were classified according to their effect size values by determining their asset values. User profiles were classified into three different categories, namely popular-active, observer-passive and spam-bot-malicious by using K-Nearest Neighbor (K-NN), Support Vector Machine(SVM) and Artificial Neural Network (ANN) algorithms. For classification, the study used the basic characteristics of users, such as density, centralization and diameter, as well as suggested time series such as the Simple Moving Average and Cumulative Moving Average. The highest accuracy was obtained by the K-NN algorithm. The results obtained with K-NN for all classes are higher than the F1-Score values obtained for the other algorithms. According to the results obtained, classification accuracy values were found to reach a maximum of 96.81% and a minimum of 92.33%. Our classification results showed that the proposed method was satisfactory for Popular-Active, Observer-Passive and Spam-Bot-Malicious account separation.
The introduction is comprised of too many general statements, with a proper backing in references. For example, lines 31- 33: the number of social media users has started increasing exponentially, thereby changing communication platforms, socializationareas and topics. Is there an official sources for this increase, a period of reference/a certain place, in what way communication platforms were changed ? etc. You may want to consult the following: Boulianne, S. (2015). Social media use and participation: A meta-analysis of current research. Information, communication & society, 18(5), 524-538.
The following references have been added to the article.
Boulianne, S. (2015). Social media use and participation: A meta-analysis of current research. Information, communication & society, 18(5), 524-538.
https://www.smartinsights.com/social-media-marketing/social-media-strategy/new-global-social-media-research/
The same applies to many other phrases:
“This change causes highly effective transformations, especially in economic, political and social areas”. What do you mean by “highly effective transformations”?
This change causes highly effective transformations, especially in economic, political and social areas such as the sudden rise/fall in stock exchange and the determination of government policies.
Lines 36-38: “People prefer to use social media to make contact with their environment and get faster and more effective returns; similarly, companies use it to make contact with their customers, institutions and organizations within their target audiences”. Can you cite a source acknowledging these motivations for social media use or do you take it as common knowledge?
We take it as common knowledge.
Lines 47-48: the role of twitter in the Arab Spring, a source for assessing its “determining role”. See: Arafa, M., & Armstrong, C. (2016). " Facebook to Mobilize, Twitter to Coordinate Protests, and YouTube to Tell the World": New Media, Cyberactivism, and the Arab Spring. Journal of Global Initiatives: Policy, Pedagogy, Perspective, 10(1), 6.
The following article is added to references.
Arafa, M., & Armstrong, C. (2016). " Facebook to Mobilize, Twitter to Coordinate Protests, and YouTube to Tell the World": New Media, Cyberactivism, and the Arab Spring. Journal of Global Initiatives: Policy, Pedagogy, Perspective, 10(1), 6.
This reference supports the following description.
" It played a determining role in social events such as the Arab Spring, where demonstrators organized and made their voices heard [3]. "
Some phrases are so general that seem to be even a truism: lines 60-61 - Consistent, real and comprehensive data is essential for effective knowledge discovery.
In knowledge discovery, the fact that the data set is up to date, comprehensive and consistent plays an important role in the performance of the study. With this in mind, the following statement was used.
Consistent, real and comprehensive data is essential for effective knowledge discovery.
At a deeper level, there is a very abrupt transition to the idea of fake news, a concept used in rather confuse manner next to “fake profiles” and “fake accounts”. Are they interchangeable?
Thank you for the reviwer's suggestion. The Reviwer proposal was not fully understood by us.
Fake news is closely associated with fake profiles. Therefore, the presentation of the article was not changed.
Line 73 – typo - contributions. Listed in the current form, this subsection should be moved at the end of paper, in the discussion/conclusions part. In the introduction you may want to rephrase some of these ideas, so that you it becomes clearer how this paper addresses the gap in the literature and what are the research questions.
Discussion section has been added to the article. The contribution section has been moved to the discussion section.
Discussion
One of the major problems in social media platforms such as Twitter is bot, Sybill, fake, spam accounts, which are controlled by automated software, which are often used for malicious activities. It is important to identify Bot, Sybill, fake, spam accounts that affect a community on a particular topic, spread false information, direct people to illegal organizations, manipulate people for stock exchange transactions, and attempt to spread their private information. The determination of bot, fake, Sybill accounts has been widely studied in the literature. The studies on the separation of bot, fake and Sybill accounts from normal users by using machine learning methods are given in Table 7. With the use of many machine learning algorithms, normal user and bot, fake, Sybill account classification has been determined with 90%–96% accuracy. The studies in the literature suggested the use of one or more tweet features, periodic features and user features. In this article, the separation of Popular-Active, Observer-Passive and Spam-Bot-Malicious groups has an average accuracy of 96.81% with the help of the SVM, K-NN and ANN algorithms.
The contributions made to the literature in this article can be summarized as follows:
The users were categorized into three different categories, namely, spam-bot-malicious, popular-active, and observer-passive. Unlike existing literature, diameter, density, reciprocity, centralization and modularity parameters were used to determine users’ interactions. Simple Moving Average and Cumulative Moving Average were used with time series to separate all outlier, incoherent and inconvenient noisy data, and to complete missing data. The SVM, ANN, and K-NN classification algorithms were used and the classification accuracy values had maximum value of 96.81% and a minimum of 92.33%.
Our classification results showed that the proposed method was satisfactory for Popular-Active, Observer-Passive and Spam-Bot-Malicious account separation. The results determined that 10 different parameters were adequate and effective in classifying the users.
Table 7. Methods and Success Rates of Studies in the Literature.
|
Reference |
Method |
Accuracy |
Account detected |
|
[22] |
Random Forest, Artificial Neural Network, and Support Vector Machine |
96.3 % 96% 95.6% |
Spam |
|
[32] |
Random Forest |
86.44% |
Automated, Normal Users |
|
[33] |
Decision tree, Bayesian Network, Support Vector Machine, and Artificial Neural Network |
~90% |
Human, Sybill |
|
[34] |
Random Forest |
95% |
Bot, Sybill |
|
[35] |
Warped Correlation Method |
94% |
Bot |
|
[36] |
Logistic Regression, Naive Bayes, Support Vector Machine, and Gradient Boosted Trees |
75% 78% 82% 86% |
Bot |
|
[29] |
Naive Bayes and Entropy Minimization Discretization |
90% |
Fake |
|
[31] |
Random Forest |
96% |
Human, Bot, and Cyborg |
|
[37] |
Artificial Neural Networks, Decision Tree Classifier and Random Forest Classifier |
92.08% 94.58% 95% |
Fake |
|
Proposed Method |
Support Vector Machine, K-Nearest Neighbors and Artificial Neural Network |
96.81% |
Popular-Active, Observer-Passive and Spam-Bot-Malicious |
Line 85 – you may completely eliminate the subsection title. It is just a common paragraph at the end of the introduction that does not need a heading.
The paper organization subsection is combined with the introduction section.
The rest of the study was organized as follows. The second section presents literature studies. The third section includes the methods used and the formation of the data. The experimental results are given, and the method applied is evaluated in the fourth section. The comparison of the obtained results with the literature is given in the fifth section. The conclusion section, Section 6, outlines the contribution of this study to published literature and discusses the future development of the method applied.
I don’t understand the logic of the references in text: there is 33 in line 103, and the directly to 38 in line 116.
References have been rearranged.
Section 3 – Problem definition – is much more naturally to be integrated into the introduction.
The problem defination section is combined with the introduction section.
Section 4 should be renamed so that it becomes Data and Method, or Research Methodology or anything else that skips the “Proposed”.
The sections have been renamed.
Line 230 – typo: construction.
The construction word has been deleted.
Figure 2 should be converted into a table.
The figure.2 was converted to table.4.
The conclusions should be extended in order to clearly delineate the theoretical and practical implications of the paper.
The conclusion and discussion sections have been updated.

Round 2
Reviewer 3 Report
Thank you for addressing the comments. One final proofreading from an English native would be useful in order to refine the writing style.
Author Response
Thank you for your comments and contributions.
The article was reviewed. Corrections were made.